# Neuroendocrine Neoplasms: Identification of Novel Metabolic Circuits of Potential Diagnostic Utility

**DOI:** 10.3390/cancers13030374

**Published:** 2021-01-20

**Authors:** Beatriz Jiménez, Mei Ran Abellona U, Panagiotis Drymousis, Michael Kyriakides, Ashley K. Clift, Daniel S. K. Liu, Eleanor Rees, Elaine Holmes, Jeremy K. Nicholson, James M. Kinross, Andrea Frilling

**Affiliations:** 1Department of Surgery and Cancer, Imperial College London, Exhibition Road, London SW7 2AZ, UK; b.jimenez@imperial.ac.uk (B.J.); mei.u11@imperial.ac.uk (M.R.A.U.); Panagiotis.drymousis@nhs.net (P.D.); michael.kyriakides@imperial.ac.uk (M.K.); ashley.clift@phc.ox.ac.uk (A.K.C.); daniel.liu08@imperial.ac.uk (D.S.K.L.); e.rees@imperial.ac.uk (E.R.); j.kinross@imperial.ac.uk (J.M.K.); 2Department of Metabolism, Nutrition and Reproduction, Imperial College London, Exhibition Road, London SW7 2AZ, UK; elaine.holmes@imperial.ac.uk; 3The Australian National Phenome Centre, Health Futures Institute, Murdoch University, Perth 6150, Australia; j.nicholson@imperial.ac.uk; 4Institute for Global Health Innovation, Imperial College London, Exhibition Road, London SW7 2AZ, UK

**Keywords:** neuroendocrine neoplasms, neuroendocrine tumours, biomarkers, nuclear magnetic resonance, metabolic profiling, metabonomics, precision medicine

## Abstract

**Simple Summary:**

Neuroendocrine neoplasms (NEN) are rare tumours, and currently available, mono-analyte biomarkers for diagnosis and prognosis have poor predictive and prognostic accuracy. Metabolic profiling has been applied to several cancer types, but the systemic metabolic consequences of NEN have not yet been well studied. Here, we demonstrate, in a treatment-naïve cohort of patients with NEN, that several metabolites are dysregulated in NEN and this is detectable in urine, due to changes in cancer metabolic processes, neuroendocrine signalling molecules and the gut mirobiome. This may have relevance for novel diagnostic biomarkers.

**Abstract:**

The incidence of neuroendocrine neoplasms (NEN) is increasing, but established biomarkers have poor diagnostic and prognostic accuracy. Here, we aim to define the systemic metabolic consequences of NEN and to establish the diagnostic utility of proton nuclear magnetic resonance spectroscopy (^1^H-NMR) for NEN in a prospective cohort of patients through a single-centre, prospective controlled observational study. Urine samples of 34 treatment-naïve NEN patients (median age: 59.3 years, range: 36–85): 18 had pancreatic (Pan) NEN, of which seven were functioning; 16 had small bowel (SB) NEN; 20 age- and sex-matched healthy control individuals were analysed using a 600 MHz Bruker ^1^H-NMR spectrometer. Orthogonal partial-least-squares-discriminant analysis models were able to discriminate both PanNEN and SBNEN patients from healthy control (Healthy vs. PanNEN: AUC = 0.90, Healthy vs. SBNEN: AUC = 0.90). Secondary metabolites of tryptophan, such as trigonelline and a niacin-related metabolite were also identified to be universally decreased in NEN patients, while upstream metabolites, such as kynurenine, were elevated in SBNEN. Hippurate, a gut-derived metabolite, was reduced in all patients, whereas other gut microbial co-metabolites, trimethylamine-*N*-oxide, 4-hydroxyphenylacetate and phenylacetylglutamine, were elevated in those with SBNEN. These findings suggest the existence of a new systems-based neuroendocrine circuit, regulated in part by cancer metabolism, neuroendocrine signalling molecules and gut microbial co-metabolism. Metabonomic profiling of NEN has diagnostic potential and could be used for discovering biomarkers for these tumours. These preliminary data require confirmation in a larger cohort.

## 1. Introduction

Neuroendocrine tumours, recently reclassified as neuroendocrine neoplasms (NEN), were historically regarded as rare entities and arise from the gastroenteropancreatic system in about 70% of cases. Multiple epidemiological studies have described a steadily increasing incidence [1,2] and NEN frequently present with distant metastases at the initial diagnosis [3]. Fewer than 20% of patients with distantly metastatic (stage IV) NEN are candidates for surgery with curative intent. Currently, effective treatment options for “non-surgical” patients with advanced tumours are limited, and although novel modalities have been demonstrated to improve progression-free survival in randomised clinical trials [4,5,6,7], ramifications in terms of improved overall survival are yet to be identified [8]. Standard tumour markers for neuroendocrine disease include the neurosecretory peptides chromogranins A and B (CgA, CgB) and a panel of cell-type-specific secretory products including gastrin, insulin, pancreatic polypeptide, vasoactive intestinal peptide, and serotonin or its urinary metabolites, such as 5-hydroxyindoleacetic acid (5-HIAA). Their clinical utility is burdened by limited accuracy [9] and there is an urgent unmet need for novel non-invasive biomarkers for early identification, treatment monitoring, precision diagnostics and phenotyping of NEN for the stratification of therapy.

Systems medicine provides a compelling opportunity to develop novel diagnostic and prognostic strategies for rare diseases, where a diagnosis may be made on a deeper analysis of an individual’s biology, based on the measurement of many thousands of genes, proteins or metabolites [10]. Although the role of miRNA [11,12], circulating tumour cells [13], circulating cell-free DNA and copy number variations [14] or circulating neuroendocrine gene transcripts [15,16] in NEN have been recently analysed and encouraging results reported, to date, there is no compelling “omics based” technology in routine clinical use. One of the most widely applicable aspects of the development of precision medicine relates to the diverse applications of metabolic phenotyping (metabotyping) to clinical diagnostics, prognostics and molecular epidemiology [10]. Metabolic profiling has been shown to provide promising biomarker panels in various neoplasms, including prostate cancer [17], breast cancer [18], colorectal cancer [19,20], hepatocellular carcinoma [21], lung cancer [22] and leukaemia [23]. The metabotypes of individuals can be measured from the composition of accessible biofluids or tissues sampled in the clinic. Metabotypes vary extensively between individuals and populations and result from the complex interplay of host genes, lifestyle, diet and gut microbes. The systems metabolism of NEN has not previously been described in detail and subtle metabolic perturbations induced by NEN remain poorly defined.

Metabolic phenotyping may provide novel diagnostic and prognostic strategies for various diseases [10]. Urinary analysis also delivers the greatest possible chance of achieving this, as it provides a non-invasive “window” into gastrointestinal metabolism. In our previous discovery study, we demonstrated that retrospective proton nuclear magnetic resonance (^1^H-NMR) analysis of urine was able to describe distinct metabolic phenotypes for NEN based on the clinical presentation [24]. The aim of the current study was, therefore, to define the systemic metabolic consequences of NEN and to establish the diagnostic utility of ^1^H-NMR for NEN in a prospective study of treatment-naïve patients with either pancreatic NEN (PanNEN) or small bowel NEN (SBNEN) which, together, account for the vast majority of digestive NEN.

## 2. Results

A total of 61 participants were recruited (NEN *n* = 41, of which: PanNEN *n* = 21, SBNEN *n* = 20, and controls *n* = 20). Of these, one healthy control was excluded due to corruption in the data file and seven patients were excluded from the analysis as they behaved as chemical outliers on PCA Figure 1.

One patient was excluded due to poor water suppression of the spectrum; three further spectra were dominated by a high concentration of an unknown drug metabolite which masked a considerable portion of the spectra, and three patients had spectra which could not be aligned satisfactorily due to abnormal pH and osmolality, and were hence unsuitable for analysis. In total, 34 NEN (18 PanNEN and 16 SBNEN) patients and 19 healthy controls were included in the final analysis. The median age of the included NEN patients was 59.3 years (range 36–85), and further demographic data are shown in Table 1. Of the healthy controls, nine were female, the mean age was 57.2 years (range 31–68), and the median BMI was 25.1 (range 17.9–28.7).

Seven of 18 (39%) of the PanNEN patients had functioning tumours. Of these, four were insulinoma, two gastrinoma and one glucagonoma. Twenty-seven patients (79%) had metastatic disease at initial diagnosis; of them, three out of 18 (17%) of the PanNEN patients and 11 out of 16 (69%) SBNEN presented with metastases to the liver. Five of the small bowel NEN patients had carcinoid syndrome (all untreated at study entry). The disease stage remained unchanged in 31 patients during the 3-year follow-up.

Supervised orthogonal projection to latent structures discriminant analysis (OPLS-DA) modelling of the NMR spectra revealed that the urinary metabolic phenotype of NEN was clearly differentiated from that of healthy individuals (Figure 2 and Figure 3).

Valid OPLS-DA models could be built, with permutation *p* < 0.05, comparing between control and all NEN patients (R^2^Y = 0.82, Q^2^Y = 0.55, AUROC = 0.95) and comparing control to each subgroup of NEN, respectively (Control vs. PanNEN: R^2^Y = 0.85, Q^2^Y = 0.48, AUROC = 0.90, Control vs. SBNEN: R^2^Y = 0.89, Q^2^Y = 0.47, AUROC = 0.90); Figure 2. No valid model could be built comparing PanNEN and SBNEN (Q^2^Y < 0).

Using the OPLS-DA models, signals with a significant contribution towards the model coefficients were identified, their chemical identity assigned and their relative concentration obtained for further testing of significance using Wilcoxon rank sum test (Figure 4, Table 2).

For the model comparing control and all NEN patients, trigonelline (OPLS-DA *p*FDR = 0.002, rank-sum *p*FDR = 0.012), hippurate (OPLS-DA *p*FDR = 0.044, rank-sum *p*FDR = 0.132) and a niacin-related metabolite (OPLS-DA *p*FDR = 0.008, rank-sum *p*FDR < 0.001) were found to be significantly lower in NEN patients, while an *S*-methyl-L-cysteine sulfoxide-related metabolite, which is a marker of cruciferous vegetable consumption [28], was significantly higher in the NEN cohort (OPLS-DA *p*FDR = 0.02, rank-sum *p*FDR = 0.007).

Similarly, between control and PanNEN, patients had significantly lower amounts of trigonelline (OPLS-DA *p*FDR = 0.012, rank-sum *p*FDR = 0.111), hippurate (OPLS-DA *p*FDR = 0.010, rank-sum *p*FDR = 0.123) and the niacin-related metabolite (OPLS-DA *p*FDR = 0.08, rank-sum *p*FDR = 0.003) in their urine. Conversely, excretion of 2-hydroxyisobutyrate was significantly higher in PanNEN patients (OPLS-DA *p*FDR = 0.017, rank-sum *p*FDR = 0.141).

In the OPLS-DA model-comparing control and SBNEN, trigonelline and the niacin-related metabolite were also significantly reduced in patients (OPLS-DA *p*FDR = 0.053 and 0.03, rank-sum *p*FDR = 0.010 and 0.001, respectively). It also revealed that trimethylamine *N*-oxide (TMAO), 4-hydroxyphenylacetate and phenylacetylglutamine (PAG) and kynurenine were significantly increased in SBNEN patients (OPLS-DA *p*FDR = 0.002, 0.015, 0.002 and 0.003; rank-sum *p*FDR = 0.036, 0.167, 0.171 and 0.132, respectively).

5-hydroxyindole acetic acid (5HIAA) was higher in all NEN patients compared to healthy control; however, the difference did not reach significance, probably due its low levels in PanNEN Figure 4. However, 5-HIAA was significantly elevated in SBNEN patients (of whom 11/16 had liver metastases) based on the non-parametric rank sum test (rank-sum *p*FDR = 0.015). Furthermore, univariate comparisons of metabolite levels between PanNEN and SBNEN revealed that 5-HIAA was the only metabolite that significantly differed between the two NEN types’ rank-sum *p*FDR = 0.003, Table 2.

As determining the presence of metastases is a major clinical challenge of significant importance, we sought to further investigate whether the metabolic profiles of urine could be utilised to delineate NEN patients with or without metastases. Only the PanNEN group had sufficient numbers for comparison between any loco-regional and/or distant metastasis being present (*n* = 8) or absent (*n* = 10). A cross-validated PCA model was able to separate those with or without metastasis in the first principal component with high accuracy Figure 5.

Interestingly, all three patients without metastases at the time of urine collection that clustered close to spectra from patients with metastases on the negative side of principal component 1 developed recurrent disease during the 3-year follow-up post-resection. However, although a strong trend was observed in the PCA scores plot, no valid OPLS-DA models could be built to differentiate patients with metastatic from non-metastatic tumours (Q^2^Y < 0), probably owing to the small sample size.

## 3. Discussion

This is the first prospective study defining the systemic metabolic consequences of NEN and assessing metabolic phenotyping as a diagnostic tool in neuroendocrine neoplasia disease. This suggests that ^1^H-NMR has tangible clinical utility, as it is able to identify NEN status when compared to a healthy control group with a high diagnostic accuracy, and to provide candidate biomarkers, new diagnostic strategies and phenotyping for the stratification of therapy and identify therapeutic targets that could be pursued. In light of the imminent need for novel biomarkers in NEN, metabolic phenotyping deserves further attention. Chromogranin A, as a global biomarker for NEN that is currently widely used in clinical practice, is burdened by poor assay reproducibility, low sensitivity and limited predictive value. Chromogranin levels can be influenced by various medical conditions such as renal failure, inflammatory bowel disease and irritable bowel syndrome, non-NEN malignancies, and therapy with proton pump inhibitors. As expected, this analysis identified that 5-HIAA, an established biomarker currently used in the clinical diagnosis of NEN, was a discriminant between SBNEN patients and healthy controls. Of note, 68.8% of patients had liver metastases. Apart from NEN, elevation of urinary 5-HIAA has been reported in several types of malignant conditions including gastric cancer [29] and breast cancer [30]. The prognostic value of 5-HIAA has been suggested for advanced disease in SBNEN [31]. Our multivariate models showed that 5-HIAA contributed only minimal diagnostic strength to the model, but was significant in univariate non-parametric comparisons. Its level was raised in SBNEN rank-sum *p*FDR = 0.015), and it was the only metabolite found to be significantly different between PanNEN and SBNEN (rank-sum *p*FDR = 0.003; Table 2. This might be due to the higher rate of liver metastases in SBNEN in our cohort.

Similar to 5-HIAA, which is a urinary metabolite of tryptophan-derived serotonin, a number of other discriminatory metabolites identified in this study are also products of tryptophan metabolism. A niacin-related metabolite and trigonelline were found to be significantly reduced in NEN patients (OPLS-DA *p*FDR = 0.002, rank-sum *p*FDR = 0.012) and the significance remained even in comparisons to the subgroups (PanNEN: OPLS-DA *p*FDR = 0.012, rank-sum *p*FDR = 0.111 and SBNEN: OPLS-DA *p*FDR = 0.053, rank-sum *p*FDR = 0.010). Niacin (vitamin B3) and trigonelline are downstream products of tryptophan metabolism via the kynurenine pathway. Trigonelline is found in coffee and other dietary products [32], and has been shown to have anti-tumourogenic effects [33]. Liao et al. demonstrated that trigonelline inhibited migration of Hep3B cells through down-regulation of nuclear factor E2-related factor-2-dependent antioxidant enzymes’ activity [34]. However, despite both metabolites being potentially confounded by dietary factors, niacin deficiency is known to be prevalent in patients with serotonin-producing NEN [35]. In the series reported by Bouma et al., the urinary niacin metabolite N1-methylnicotinamide was reduced compared to healthy controls (median 17.9 µmol/24 h vs. 43.7 µmol/24 h, respectively) and below normal in 45% of the patients [35]. Therefore, it is possible that the reduced excretion of trigonelline and the niacin-related metabolite observed in NEN patients here is directly related to the presence of NEN and that niacin could play a role in the prevention of tumour progression.

Interestingly, kynurenine, a metabolite found upstream of the two previously discussed metabolites in tryptophan metabolism, was found to be elevated in the SBNEN group in this study. It has been increasingly recognised that indoleamine-2,3-dioxygenase (IDO) and tryptophan-2,3-dioxygenase (TDO), which catabolise tryptophan into kynurenine, is commonly upregulated in tumours and this is thought to facilitate tumour immune evasion [36]. Elevated levels of kynurenine and kynurenine/tryptophan ratio has been reported in the serum of breast [37] and lung [38] cancer patients, as well as in the urine of patients with bladder [39] and prostate [40] cancers. Our observation suggests that tryptophan metabolism is universally perturbed in NEN patients but the dynamics of the perturbation may differ between the NEN subgroups.

Urinary excretion of hippurate, which was found to be lowered in NEN patients (OPLS-DA *p*FDR = 0.044, rank-sum *p*FDR = 0.132), is modulated by the microbiota, and it derives from bacterial activity in the distal small intestine rather than in the colon [41]. Increased urinary excretion of hippurate has been associated with weight loss or with lean body mass [42]. However, there was no significant difference in BMI between the NEN patients and healthy controls. It has been repeatedly reported to be reduced in NEN patients [24], and also in various cancer types, such as renal cell carcinoma [43] and colorectal cancer [44]. This further supports the association of gut microbial dysbiosis with cancer development.

A few of the markers were found to have significantly increased levels in SBNEN patients compared to control, namely, TMAO, PAG and 4-hydroxyphenylacetate, also of gut microbial origin. TMAO has been associated with several diseases including colon cancer [45] and cardiovascular disease [46]. Previous data have shown that TMAO levels increase with the consumption of L-carnitine, which is metabolised by the gut microbiota such as Peptostreptococcaceae and Clostridiaceae families, into trimethylamine (TMA). TMA is further metabolised into TMAO by flavin monooxygenases in the liver [47]. Similarly, PAG and 4-hydroxyphenylacetate are also gut microbial co-metabolites and elevated urinary excretion of them has been reported in colorectal [19] and gastric [48] cancers, respectively. In summary, this analysis provides new evidence to suggest that host-microbial co-metabolic pathways are perturbed in both PanNEN and SBNEN, and may have diagnostic value.

The PCA model of PanNEN patients showed an observable differential segregation of patients with or without metastasis. Interestingly, all three PanNEN patients without distant metastases at the time of analysis that clustered close to spectra from patients with metastases on the negative side of principal component 1 developed recurrent disease during the 3-year follow-up. This highlights the potential for metabolic phenotyping to detect metastases earlier and more accurately than current clinical practice allows. However, the patient numbers in this study were not large enough to enable robust statistical inference. Future studies should include an increased number of patients with metastases as one of the aims and recruit accordingly to obtain sufficient numbers for comparison.

There are limitations to this study. Firstly, we have analysed a highly selected group of NEN patients and it remains a matter of further study to validate the results in the heterogeneous patient population seen in clinical practice. Our inclusion criteria in this initial study on metabolic circuits in NEN were very strict, since we wanted to control cofounding factors affecting the variables being studied and avoid false discovery results. The patient numbers in our study are small, which did not permit the correlation of the spectroscopic data with clinical parameters such as tumour grade, tumour stage and functional status or comparison with standard tumour markers. However, this will always be a challenge when prospectively studying uncommon tumours such as NEN. The small study size also prevented having a separate set of data for validation of the findings, which shall be addressed in our future work. We were also unable to definitively account for potential confounders such as diet in this work, since some of the metabolites, such as the SMCSO-related metabolite, were possibly of dietary origin. This analysis also did not set out to study the microbiome. As a result, there were no data from the luminal microbiome to corroborate the metabolic dataset, which strongly indicates disease-associated dysbiosis based on the perturbation of gut microbial (co-)metabolites such as hippurate and PAG. This will be the subject of future work. We had to exclude some patient samples because of technical reasons. We will attempt to increase the robustness of methodology to confirm the diagnostic capability of this tumour marker in clinical practice. Furthermore, the majority of SBNEN patients and many PanNEN patients have present metastasis, which may pose a challenge to identifying biomarkers for early diagnosis before the tumour has metastasised. However, this is as much a research challenge as a clinical challenge, as patients often present late with metastasis at diagnosis. Finally, ^1^H-NMR spectroscopy accesses only a portion of the metabolome, with a bias towards high-concentration metabolites. The failure rate of 10% is related to the experimental nature of the analysis and the challenges of sample storage and volume. ^1^H-NMR is a scalable, high-throughout technology that could be leveraged for translational applications. However, it is likely that this work would lead to the development of targeted, quantitative assays for the analysis of the candidate biomarkers that are described, which could be deployed at the point of care. Further studies would be necessary to provide more mechanistic insight into the biology of the metabolic perturbations caused by the presence of NEN in the organism. New studies would obviously require the analysis of larger cohorts, and targeted analytical techniques which should include the characterisation of the NEN gut microbiome. Studies which additionally acquire data from complementary techniques such as liquid-chromatography mass spectrometry, and comparison with other novel biomarkers such as circulating neuroendocrine gene transcripts (NETest) should aid metabolic profiling characterisation and delineate its clinical utility.

It is not possible to delineate cause and effect for the metabolites identified here, and we can only speculate, at this stage, as to their roles. However, we have identified an important avenue of research for the functional NEN microbiome that could be capitalised on to improve current clinical practice.

## 4. Materials and Methods

### 4.1. Patient Recruitment and Sample Collection

From January 2011 to December 2013, consecutive patients with either PanNEN or SBNEN were recruited from the European Neuroendocrine Tumor Society (ENETS) Centre of Excellence at Imperial College London NHS Healthcare Trust, UK. Data were recorded in our prospectively maintained database for NEN. Only treatment-naïve patients with a primary tumour still in place and confirmed diagnosis of a localised or metastasised NEN and a follow-up of at least 3 years were included. Patients under the age of 18, those who were pregnant, and patients undergoing NEN-specific systemic treatment (e.g., somatostatin analogues, mTOR inhibitors, chemotherapy or peptide receptor radionuclide therapy), and patients who had any previous systemic, liver directed or surgical NEN-specific treatment, were excluded. In all patients, the diagnosis of NEN was confirmed based on conventional histology and immunohistochemistry for NEN-specific markers utilising either surgical specimens or biopsy material. Tumour functional activity was defined by consideration of clinical symptoms and results of standard biochemical testing. All patients underwent staging and grading according to World Health Organization/European Neuroendocrine Tumor Society (WHO/ENETS) [25,26] and American Joint Committee on Cancer/Union for International Cancer Control (AJCC/UICC) [27] criteria. Staging included computed tomography (CT), magnetic resonance imaging (MRI), somatostatin receptor-targeted positron emission tomography (PET) PET/CT, and other diagnostic modalities tailored to individual clinical situations. Standard biochemical work-up comprised assessment of gut hormones and chromogranin A and B in serum (Imperial Supra-Regional Assay Service radioimmunoassay, London, UK) and 5-hydroxyindolacetic acid (5-HIAA) in 24 h urine (Chromsystems Instruments & Chemicals GmbH, Grafelfing, Germany).

Prospective data collection on medical history and clinical variables (NEN phenotype, disease stage and grade, standard NEN biomarkers) was performed by a single researcher (PD). Age-, sex- and body mass index (BMI)-matched healthy control individuals were recruited from a healthy population of volunteers at the same institution and at The Welcome Trust, London, UK. Patients who were on proton pump inhibitors interrupted treatment for 2 weeks prior to sample collection when clinically justifiable. The project was designed and carried out in accordance with reporting recommendations for tumour marker prognostic studies REMARK; Figure 1 and [49].

### 4.2. Sample Preparation

Patients and healthy controls were asked to provide a single sample of urine for analysis, which was collected and stored at −80 °C. Samples were prepared according to our previously published protocol [50]. In short, samples were thawed on the day of analysis. An aliquot of 600 μL of urine was placed into 2 mL Eppendorf tubes and centrifuged for 5 min at 6× *g*. A volume of 540 μL of the supernatant was transferred into a new Eppendorf tube and mixed with 60 μL of standard urine buffer [50] containing 1.5 M potassium dihydrogen phosphate at pH 7.4 in deuterium oxide, with 3-trimethyl-silyl-[2,2,3,3–^2^H_4_]propionic acid (TSP) as reference, and sodium azide to avoid bacterial growth. Eppendorf tubes were vortexed and 550 μL of the solution was placed into SampleJet 5 mm NMR tubes (Bruker BioSpin Ltd., Rheinstetten, Germany). SampleJet racks were immediately transferred to the SampleJet robot (Bruker BioSpin Ltd., Rheinstetten, Germany) and were maintained at 4 °C until measurement.

### 4.3. ^1^H-NMR Spectroscopic Analysis of Urine Samples

A standard NMR experiment for urine profiling as defined by the Clinical Phenotyping Centre (CPC) was undertaken for each urine sample [50]. The experimental set-up followed the strict protocol of the CPC and consisted of temperature calibration, water suppression optimisation, quantitation calibration and optimisation of each run for a quality control (QC) urine sample [50]. An ^1^H 1-dimensional (1D) profile was acquired for each urine sample using a standard 1D pulse sequence employing the first increment of a Nuclear Overhauser Effect pulse sequence to achieve pre-saturation of the water resonance in a Bruker 600 Avance III NMR spectrometer (Bruker BioSpin Ltd., Rheinstetten, Germany). The machine was equipped with a SampleJet robot, a 5 mm probe with high-degree *Z* gradients and an automatic tuning and matching unit. The ^1^H-NMR spectra were processed, phased, baseline corrected and calibrated in automation using TopSpin 3.2 (Bruker BioSpin Ltd., Rheinstetten, Germany). Any samples that did not conform to accepted criteria for line-width, baseline and water suppression were reacquired. In addition to the 1D NMR profile, a 2-dimensional (2D) J-res experiment was also acquired to exploit the structural properties and help with biomarker identification. The acquisition and processing parameters were as previously described elsewhere [50]. To facilitate the identification of metabolites that were determined to be of statistical importance in characterising NEN, 1D titration experiments with the corresponding chemical standard and 2D experiments such as COSY, TOCSY [51] and ^1^H-^13^C-HSQC were acquired for selected samples [52,53].

### 4.4. Statistical Analysis of the Spectral Data

All data pre-processing and statistical analyses were performed using Matlab R2016a (MathWorks, Natick, MA, USA). The spectra were imported into Matlab and digitised into 65,536 datapoints with segment widths of 0.0002 ppm using in-house scripts. Spectral regions corresponding to the TSP reference peak (<0.1 ppm), methanol (3.35–3.38 ppm), water (4.7–4.9 ppm) and urea (5.6–6.0 ppm) signal regions, and background noise in regions without signals (>9.4 ppm) were removed. The spectra were then aligned [54] and normalised [55]. The probabilistic quotient normalization based on the calculation of a most probable dilution factor by looking at the distribution of the quotients of the amplitudes of a test spectrum by those of a reference spectrum was used.

Spectra were first subjected to principal component analysis (PCA) for the identification of outliers. The original spectra of putative outliers, determined by being outside of the 95% Hotelling’s T^2^ confidence interval in the PCA scores space, were examined. Spectra were excluded from subsequent analyses as an outlier only if they contained extreme concentrations of certain drugs since drug metabolites can potentially confound or bias the models. Cross-validated PCA models were made using the leave-one-out method.

For orthogonal projection to latent structures-discriminant analysis (OPLS-DA) [56], 7-fold cross-validation was used and predictive models were further validated by 1000 permutations of the outcome vector. Integrals of individual peaks of interest were obtained and further tested by Wilcoxon rank sum test for significant difference between groups. *P*-values were adjusted for multiple testing using the Benjamini–Hochberg procedure [57] with cut-off for significance at the false discovery rate of 0.5%. Predictions from the OPLS-DA models were used for creating receiver operating characteristics (ROC) curves [58]. The investigators Mei Ran Abellona U, Beatriz Jiménez and Michael Kyriakides were blinded to clinical details.

Metabolite assignments were facilitated by in-house database comparison, statistical tools such as statistical total correlation spectrometry (STOCSY) [59], with subset optimisation by reference matching (STORM) [60], and 2-dimentional NMR data acquired from selected samples. Of the metabolites identified (*n* = 250) only those with discriminatory value were used for final analysis.

## 5. Conclusions

These findings suggest the existence of a new systems-based neuroendocrine circuit, regulated in part by cancer metabolism, neuroendocrine signalling molecules and gut microbial co-metabolism. This may represent a novel avenue for the discovery of precisional medicine for NEN. The metabolic profiling of NEN has diagnostic potential and should be expanded to larger studies to develop next-generation assays for precision NEN phenotyping that assesses the levels of activity of these pathways. The role of the gut microbial changes in the aetiology of NEN now needs to be defined based on the dysbiosis observed in this patient population.

## Figures and Tables

**Figure 1 cancers-13-00374-f001:**
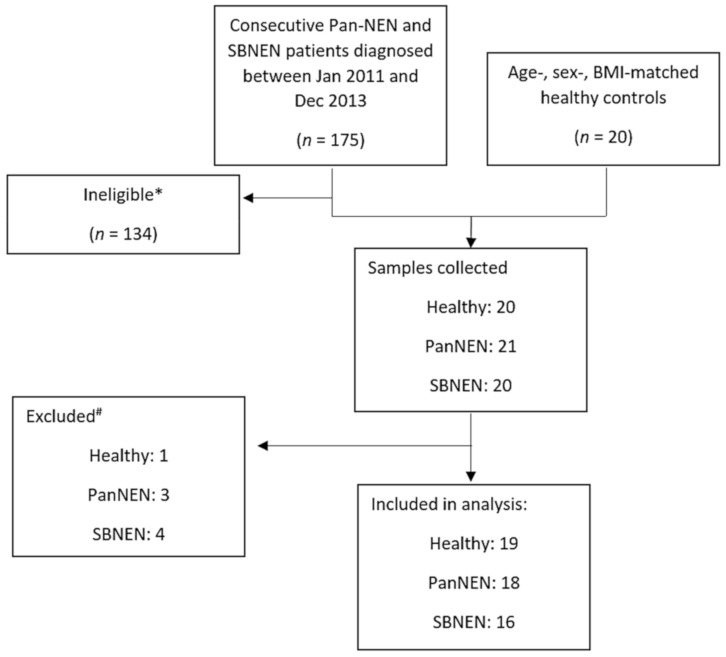
Remark diagram of the cohort studied.* Exclusion criteria are: patients under the age of 18, pregnancy, patients undergoing neuroendocrine neoplasm (NEN)-specific systemic treatment, second malignancies, comorbidities requiring significant systemic treatment (e.g., immunosuppression), impaired renal function, poor compliance, missing consent for study participation. ^#^ 1 healthy control sample due to the corruption of the original data file; for NEN samples, 1 due to poor water suppression, 3 due to high intensity of an unknown drug signal, and 3 due to extreme misalignment of the spectra. PanNEN = pancreatic neuroendocrine neoplasm, SBNEN = small bowel neuroendocrine neoplasm, BMI = body mass index.

**Figure 2 cancers-13-00374-f002:**
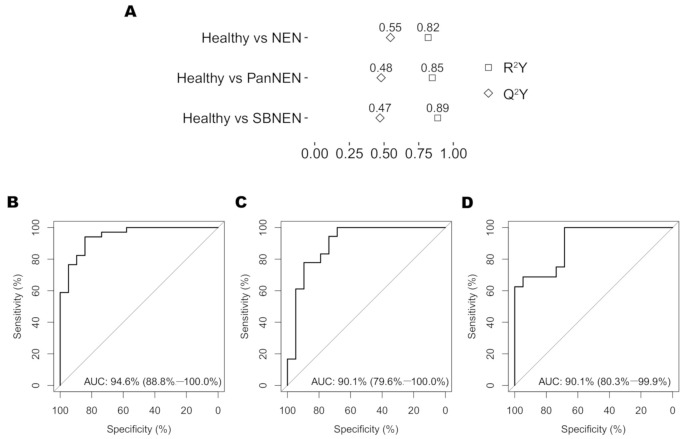
Ability of the OPLS-DA models to predict NEN status from healthy controls. (**A**) R^2^Y and Q^2^Y coefficients of the OPLS-DA models generated. (**B**) Receiver operating curves (ROC) and their associated area under the receiving operator curve (AUROC) values for Healthy vs. NEN (AUROC 0.94), (**C**) Healthy vs. PanNEN (AUROC 0.90) and (**D**) Healthy vs. SBNEN (AUROC 0.90). These demonstrate strong diagnostic utilities of the models for NEN and its subgroups.

**Figure 3 cancers-13-00374-f003:**
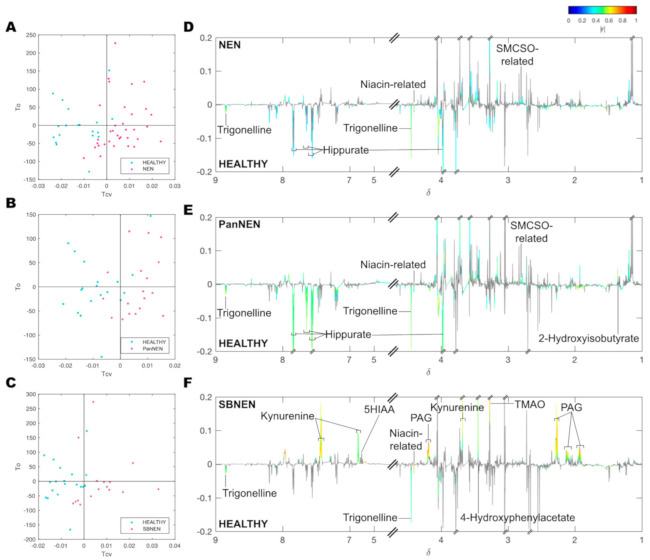
OPLS-DA models comparing NEN and its subgroups to healthy control samples. Scores (**A**–**C**) and their respective loadings (**D**–**F**) plots of the model comparing healthy individuals versus all NEN patients (**A**,**D**), healthy individuals versus those with PanNEN (**B**,**E**) and healthy individuals versus those with SBNEN (**C**,**F**). The pseudospectra of the loadings plots are coloured according to correlation coefficient |*r*| for peak positions where positive false discovery rate (*p*FDR) < 0.1.

**Figure 4 cancers-13-00374-f004:**
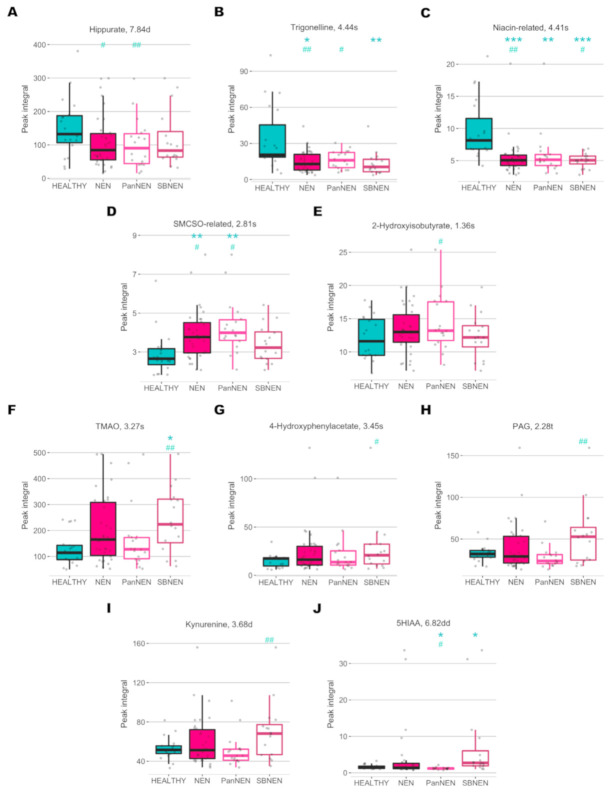
Boxplot of metabolites with significant differences between NEN and NEN subgroups compared to healthy control: (**A**) Hippurate, (**B**) Trigonelline, (**C**) metabolite related to niacin, (**D**) S-methyl-L-cysteine sulfoxide, SMCSO, related metabolite, (**E**) 2-Hydroxybutyrate, (**F**) Trimethylamine N-oxide, TMAO, (**G**) 4-Hydroxyphenylacetate, (**H**) Phenylacetylglutamine, PAG, (**I**) Kynurenine, (**J**) 5-Hydroxyindoleacetic acid, 5HIAA. Numbers in the title correspond to the chemical shift in ppm of the metabolite signal used for integration and the letters describe the signal multiplicity: singlet, s, doublet, d, doublet, dd, triplet, t The symbols * and # indicate the levels of significance compared to healthy control based on Wilcoxon’s rank sum test (*) and OPLS-DA model correlation (#), respectively: * and # for *p*FDR < 0.05; ** and ## for *p*FDR < 0.01; *** for *p*FDR < 0.001.

**Figure 5 cancers-13-00374-f005:**
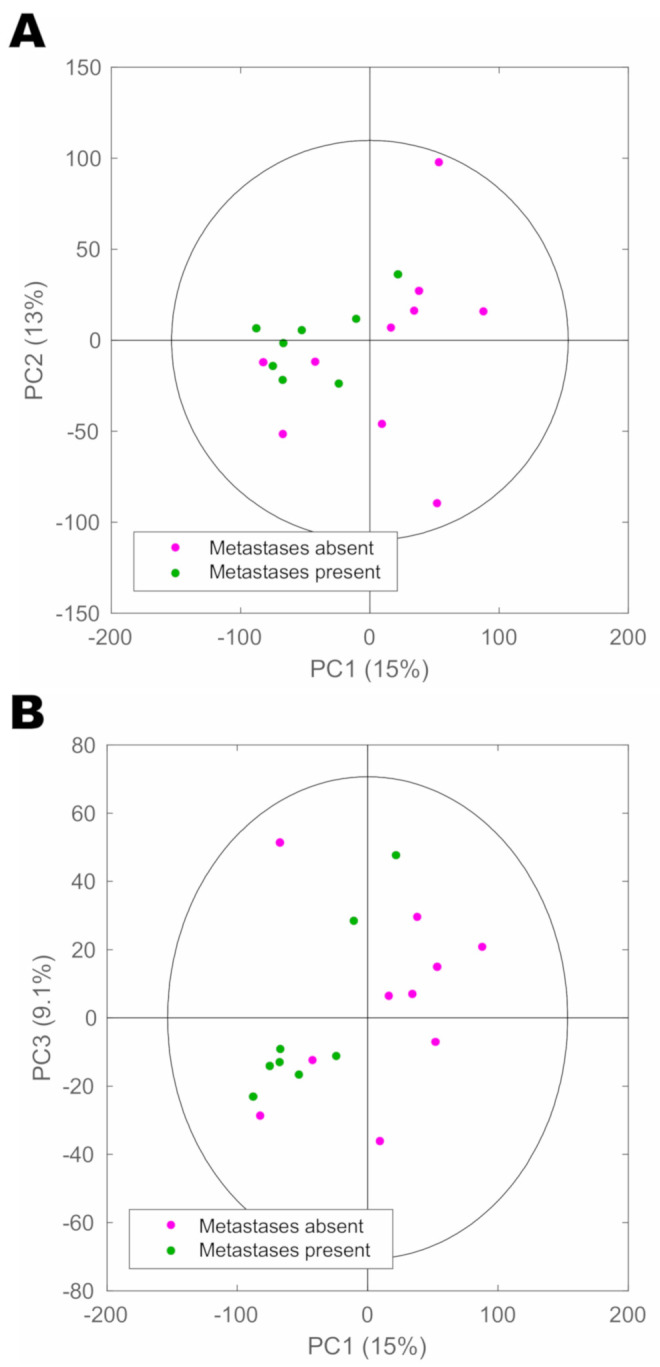
Scores plot of cross-validated principal component analysis model of spectra from PanNEN patients with and without any metastases. (**A**) Component 1 vs. 2; (**B**) Component 1 vs. 3.

**Table 1 cancers-13-00374-t001:** Clinicopathological characteristics of neuroendocrine neoplasm patients at study entry.

Parameter	Number
Number of patients	34
Sex	-
Male	21
Female	13
Ethnicity	-
Caucasian	20
African/Caribbean	5
Asian/Arabic	9
Median age at diagnosis (years)	59.3 (range 36–85)
Median BMI	27.8 (range 17.4–42.0)
Site of primary tumour	-
Small Bowel	16
Pancreas (sporadic)	18
Tumour functionality–Pancreas	-
Non-functioning	11
Functioning	7
Median serum chromogranin A (normal < 60 pmol/L)	42 (range 21–2342)
Median 5-HIAA in 24 h urine (normal 0.0–45.0 µmol/L)	25.5 (range 9.4–581.4)
Tumour Grade *	-
1	18
2	16
3	0
Tumour stage ^#^	-
T_1–4_N_0_M_0_	7
T_1–4_N_1_M_0_	13
T_1–4_N_0_M_1_	2
T_1–4_N_1_M_1_	12
Liver metastases present	-
Pancreas NEN	-
Yes	3
No	15
Small bowel NEN	-
Yes	11
No	5

BMI = body mass index; * = tumour grading performed in accordance with World Health Organization/European Neuroendocrine Tumor Society grading system [25,26], ^#^ = staging in accordance with the American Joint Committee on Cancer/Union for International Cancer Control staging system [27], NEN = neuroendocrine neoplasm.

**Table 2 cancers-13-00374-t002:** Correlation coefficients and *p*-values of chemical shifts from OPLS models and Wilcoxon’s rank sum test.

OPLS-DA Model									Wilcoxon’s Rank Sum Test							
-			Healthy Vs. NEN		Healthy Vs. PanNEN		Healthy Vs. SBNEN		Healthy Vs. NEN		Healthy Vs. PanNEN		Healthy Vs. SBNEN		PanNEN Vs. SBNEN	
Metabolite	ppm	Multiplicity	*r*	*p*FDR	*r*	*p*FDR	*r*	*p*FDR	*p*	*p*FDR	*p*	*p*FDR	*p*	*p*FDR	*p*	*p*FDR
Hippurate	3.978	s	-	0.199	−0.488	0.022	-	0.867	0.105	0.173	0.075	0.123	0.345	0.423	0.438	0.751
	7.557	t	−0.318	0.098	−0.511	0.016	-	0.642	0.061	0.129	0.068	0.123	0.18	0.263	0.569	0.751
	7.644	t	-	0.107	−0.511	0.016	-	0.681	0.071	0.132	0.075	0.123	0.202	0.275	0.523	0.751
	7.84 *	d	−0.375	0.044	−0.549	0.01	-	0.488	0.073	0.132	0.081	0.123	0.202	0.275	0.666	0.751
Niacin-related	4.405 *	s	−0.46	0.008	−0.391	0.08	−0.517	0.03	0	0	0	0.003	0	0.001	0.666	0.751
	8.791	d	−0.342	0.072	-	0.213	-	0.156	0.01	0.035	0.075	0.123	0.008	0.029	0.641	0.751
Trigonelline	4.444 *	s	−0.525	0.002	−0.533	0.012	-	0.053	0.003	0.012	0.045	0.111	0.002	0.01	0.116	0.295
	8.85	m	−0.457	0.009	−0.411	0.063	−0.509	0.034	0.003	0.012	0.047	0.111	0.002	0.01	0.081	0.238
	9.128	s	−0.485	0.005	−0.508	0.017	−0.462	0.067	0.005	0.019	0.092	0.129	0.002	0.01	0.037	0.156
2-Hydroxyisobutyrate	1.363 *	s	-	0.125	0.507	0.017	-	0.815	0.312	0.439	0.104	0.141	0.987	0.987	0.221	0.441
PAG	1.935	m	-	0.299	-	0.687	0.648	0.002	0.788	0.907	0.218	0.252	0.066	0.132	0.028	0.154
	2.107	m	-	0.886	−0.48	0.025	0.553	0.017	0.897	0.921	0.062	0.123	0.074	0.133	0.016	0.115
	2.276 *	t	-	0.537	-	0.092	0.644	0.002	0.704	0.836	0.035	0.101	0.108	0.171	0.016	0.115
	4.187	m	-	0.307	-	0.49	0.726	0	0.623	0.789	0.013	0.046	0.071	0.133	0.009	0.115
SMCSO-related	2.809	s	0.42	0.02	0.454	0.036	-	0.162	0.001	0.007	0	0.003	0.066	0.132	0.056	0.192
4-Hydroxyphenylacetate	3.446 *	s	-	0.129	-	0.432	0.557	0.015	0.228	0.333	0.649	0.685	0.101	0.167	0.438	0.751
TMAO	3.273 *	s	-	0.465	-	0.998	0.653	0.002	0.065	0.13	0.494	0.537	0.011	0.036	0.051	0.192
Kynurenine	3.679 *	d	-	0.179	-	0.864	0.626	0.003	0.853	0.921	0.171	0.204	0.066	0.132	0.018	0.115
	6.87	dd	-	0.328	-	0.994	0.55	0.018	0.889	0.921	0.081	0.123	0.03	0.071	0.01	0.115
	7.428	t	-	0.388	-	0.246	0.646	0.002	0.817	0.913	0.23	0.257	0.082	0.142	0.034	0.156
5HIAA	6.819 *	dd	-	0.48	−0.482	0.024	-	0.151	0.985	0.985	0.008	0.036	0.003	0.015	0	0.003

Correlation coefficients of only those with positive false discovery rate (*p*FDR) < 0.05 shown. * peak with the greatest signal-to-noise ratio and/or least overlap that was used for interpretation and reporting. Multiplicity: s = singlet, d = doublet, t = triplet, dd = doublet of doublet, m = multiplet. 5-HIAA: 5 Hydroxyindolacetic acid; PAG: phenylacetylglutamine; OPLS-DA: Orthogonal partial least squares-discriminant analyses; SMCSO: *S*-methyl-L-cysteine sulfoxide; TMAO: Trimethylamine-*N*-oxide.

## Data Availability

The data presented in this study are available on request from the corresponding author. The data are not publicly available as they are derived from sensitive medical information from individuals with neuroendocrine neoplasms or are study ‘controls’. Therefore, ethical approval for data sharing would need to be sought.

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
