# Peer review of "Neuroendocrine Neoplasms: Identification of Novel Metabolic Circuits of Potential Diagnostic Utility"

_cancers, 2021, doi:10.3390/cancers13030374_

Round 1
Reviewer 1 Report
Jimenez, Mei Ran et al. present prospectively-collected urinary metabolic profiles of untreated neuroendocrine neoplasm (NEN) patients. Their research is enticing, novel and highly relevant, warranting publication. However, few minor comments remain:
General remarks:
- The tryptophan metabolism has been frequently implicated in small bowel NENs (where serotonin producing tumors are frequently seen). However, the role of thryptophan metabolism in pancreatic NENs is far less studied (as they typically don’t excrete serotonin in quantities leading to clinical symptoms). The downregulation of niacin and trigonelline in in both small bowel and pancreatic NENs is therefore highly interesting and should be discussed this in more detail.
- Before this technique can be implemented in clinical practice, the technical failure rate (>10% ) should be reduced. It might be interesting to discuss which strategies could be used to reduce this failure rate.
Specific remarks
- Line 59: Recently some date on circulating tumor DNA has been published by Boons et al, Front. Oncol., 01 November 2018, which could be interesting as potential biomarker (next to metabolic profiling).
- Line 101: Can you specify the type of functioning tumor that was seen in the pancreatic NENs? Did you also include patients with small bowel NEN and carcinoid syndrome and, if so, how many?
- Line 164: Why did you only look in the pancreatic group and not in all NENs combined at the difference between metastatic and non-metastatic disease?
- Line 208: The statement on niacin supplementation and its possible role in prevention of tumor progression is rather strong and not entirely backed by the data. Please revise.
- Line 209: As the data is not part of this manuscript nor published elsewhere, I would suggest removing the clinical observation on niacin levels in blood.
- Line 296: I think it should be “when clinically justifiable” instead of “unless clinically justifiable”
Author Response
Jimenez, Mei Ran et al. present prospectively-collected urinary metabolic profiles of untreated neuroendocrine neoplasm (NEN) patients. Their research is enticing, novel and highly relevant, warranting publication. However, few minor comments remain:
General remarks:
- The tryptophan metabolism has been frequently implicated in small bowel NENs (where serotonin producing tumors are frequently seen). However, the role of tryptophan metabolism in pancreatic NENs is far less studied (as they typically don’t excrete serotonin in quantities leading to clinical symptoms). The downregulation of niacin and trigonelline in in both small bowel and pancreatic NENs is therefore highly interesting and should be discussed this in more detail.
- Before this technique can be implemented in clinical practice, the technical failure rate (>10% ) should be reduced. It might be interesting to discuss which strategies could be used to reduce this failure rate.
We would like to thank the reviewer for their overall positive assessment of our work and their very thoughtful remarks.
The failure rate of 10% is related to the experimental nature of the analysis and the challenges of sample storage and volume. 1HNMR is a scalable, high-throughput technology that could be leveraged for translational applications. It is likely that this work could lead to the development of targeted, quantitative assays for the analysis of the candidate biomarkers that are described which could be deployed at the point of care. We have added this information in the text.
We have altered the text to state that further studies would be necessary to provide more mechanistic insight into the biology of the metabolic perturbations caused by the presence of NEN tumours in the organism. New studies would obviously require the analysis of larger cohorts, and targeted analytical techniques which should include the characterisation of the NEN gut microbiome.
Specific remarks
- Line 59: Recently some date on circulating tumor DNA has been published by Boons et al, Front. Oncol., 01 November 2018, which could be interesting as potential biomarker (next to metabolic profiling).
We have added this information in the text. - Line 101: Can you specify the type of functioning tumor that was seen in the pancreatic NENs? Did you also include patients with small bowel NEN and carcinoid syndrome and, if so, how many?
Thank you for bringing this to our attention. Of the functioning PanNEN, 4 were insulinomas, 2 were gastrinomas and there was 1 glucagonoma. Five of the small bowel NEN patients had carcinoid syndrome, all untreated at the study entry (according to inclusion criteria). We have added this information in the text.
- Line 164: Why did you only look in the pancreatic group and not in all NENs combined at the difference between metastatic and non-metastatic disease?
In this initial study our aim was to analyse homogeneous groups of NEN. As it is increasingly understood, NEN are highly heterogeneous collective of tumours, and including multiple other forms of the disease may increase heterogeneity to the extent that such molecular analyses become very difficult to dissect. We wanted to focus on the clinically most relevant gastro-entero-pancreatic (GEP) NEN and have therefore chosen pancreatic NEN and small intestinal NEN which account for approximately 70% of all GEP NEN. Furthermore we felt that NEN with high tendency to metastasise would be more interesting for a study on potential novel biomarker than e.g. well-differentiated appendiceal NEN, small rectal NEN, or type 1 gastric NEN which nearly never present in stage IV. A study of this type would be virtually impossible in appendiceal NEN or small rectal NEN since they are in general detected and treated ‘incidentally’ at the time of an intervention (appendectomy and endoscopic biopsy, respectively). We absolutely agree that research in patients with other GEP NEN than those analysed in in this study would be of academic interest. As assumed, a certain specific type of diet and associated specific gut microbioma may contribute to high rates of rectal NEN seen in Asian populations.
- Line 208: The statement on niacin supplementation and its possible role in prevention of tumor progression is rather strong and not entirely backed by the data. Please revise.
We have revised the text accordingly. - Line 209: As the data is not part of this manuscript nor published elsewhere, I would suggest removing the clinical observation on niacin levels in blood.
We have revised the text accordingly. - Line 296: I think it should be “when clinically justifiable” instead of “unless clinically justifiable”
We have revised the text accordingly.
Reviewer 2 Report
In general, this is a very complex work investigating a number of urinary metabolites in a group of pancreatic and small-bowel NENs, compared with healthy subjects.
Although the methodology used is impressive in terms of technology and number of evaluated factors, I have some concerns regarding the utility of this complex (and I suppose expensive) diagnostic approach.
After reading this sentence in the discussion (the first sentence) "1H-NMR has tangible clinical utility as it is able to identify patients with NEN when compared to healthy control group with a high diagnostic accuracy " I wonder why physicians should use 1H-NMR to identify NEN? What is its real utility? How its accuracy may be assessed compared with the "standard" diagnostic approach? What is the real potential utility of this technology in the diagnostic work-up of NEN patients?
Since I was not able to find good answers to these questions in the manuscript, I am unable to fully understand what is the meaning of this work, which in my opinion remains a charming, descriptive, well-done investigation with no clear message for the reader.
Specific comments
- aim: " to define the systemic metabolic consequences of NEN", I don't fully understand this statement
- Patients were enrolled between 2011 and 2013, I wonder why data are submitted for publication in 2020
- I don't understand why patients were selcted according with available follow-up > 3 years, I don't see data on patients' follow-up.
- There is no correlation with grading, staging, and validated tumor markers (this is also reported by the authors in the limitations of the study)
- In the discussion, I believe a comment on other novel (and more solid) diagnostic tool should be given (i.e. NETest)
- The number of healthy control is low, maybe it should be increased to better validate the model.
- Additional data on matching cases vs controls should be given (age, sex, drugs consumption)
Author Response
In general, this is a very complex work investigating a number of urinary metabolites in a group of pancreatic and small-bowel NENs, compared with healthy subjects.
Although the methodology used is impressive in terms of technology and number of evaluated factors, I have some concerns regarding the utility of this complex (and I suppose expensive) diagnostic approach.
Thank you for your positive comments on the technology used in this study. We agree that it is complex and challenging, however it is broadly available and used in a substantial number of centres worldwide for research applied to clinical diagnostics, prognostics, stratification to treatment, monitoring of treatment response, pharmacotherapy, and molecular epidemiology.
After reading this sentence in the discussion (the first sentence) "1H-NMR has tangible clinical utility as it is able to identify patients with NEN when compared to healthy control group with a high diagnostic accuracy " I wonder why physicians should use 1H-NMR to identify NEN? What is its real utility? How its accuracy may be assessed compared with the "standard" diagnostic approach? What is the real potential utility of this technology in the diagnostic work-up of NEN patients?
We are sorry to learn that we were not able to sufficiently accentuate the potential role and utility of 1H-NMR in NEN. We have rephrased the second sentence in the discussion. 1HNMR is a high throughput analytical platform that could indeed be leveraged as a translatable method for precision phenotyping of NEN through the analysis of multiple metabolic parameters. Its value proposition is that it provides a comprehensive systems overview of NEN metabolism that could be applied for clinical diagnostics but that can also be deployed for prognostics, stratification to treatment and monitoring of treatment response.
We would like to thank the reviewer for an array of thoughtful questions. The need for early diagnosis is predicated upon the delay in diagnosis of NEN, which has typically been within the range of several years, despite development of modern imaging methods. Late diagnosis restricts treatment effectiveness and the absence of adequately sensitive and specific biomarkers obviates accurate monitoring of disease progression. Most neoplasia treatment is dependent on the assessment of many pathological criteria, whereas pathological interpretation to predict disease aggressiveness in NEN is limited as it is based on overlapping histological features in a biologically diverse disease cohort. Prognostication based on pathological grading has manifold issues; e.g. in G1/G2 group Ki67 index does not discriminate between locoregional and metastasised stage. Treatment options for patients with NEN are diverse, including SSA, PRRT, immunotherapy, cytotoxic chemotherapy, targeted drugs, interventional radiological approaches, and surgery. This plethora of expensive and sometimes toxic treatment choices, usually selected empirically, highlights the need to monitor tumour responsiveness both in clinical trials and in routine practice. For most NEN, tumour responsiveness is almost entirely assessed through imaging. This strategy has obvious limitations in relation to RECIST. Present imaging modalities have great limitations to define persistent or (early) progressive disease, monitor effectiveness of treatment, and predict aggressive tumour behaviour. Furthermore, cumulative radiation exposure and costs associated with repetitive imaging in the long-term follow-up of patients who might exceed a 10-year life expectancy, supports the need for accurate biomarkers that directly measure tumour-cell activity and provide real-time feedback to the clinician. In other cancers, such as breast cancer, the development of molecular markers has advanced disease management. Such information informs clinical decision making with respect to the choice and the timing of therapy, assessment of effectiveness, and providing, in some instances, prognostic information.
In their specific comments the reviewer is referring to the NETest as a “more solid” diagnostic tool. It took more than 5 years from initial introduction and >40 publications until the NETest has been recognised as “helpful”. Still, it is not authorised by official bodies (FDA, EMA) and not yet widely used. We hope that our ongoing and future research will allow us to give the reviewer a more specific answers regarding the potential utility of 1H-NMR technology in the work-up of NEN patients.
Since I was not able to find good answers to these questions in the manuscript, I am unable to fully understand what is the meaning of this work, which in my opinion remains a charming, descriptive, well-done investigation with no clear message for the reader.
We have addressed this comment above. However, we respectfully refute the observation that this is nothing more than a ‘charming’ analysis. It provides novel insight into the metabolic networks that are perturbed in patients with NEN and in turn provides candidate biomarkers, potentially new diagnostic strategies (figure 2), a tool for monitoring of treatment response and candidate therapeutic targets that could be pursued.
Specific comments
- aim: " to define the systemic metabolic consequences of NEN", I don't fully understand this statement
This is a comprehensive analysis of the global metabolic perturbations caused by the presence of NEN and its metastases. 1HNMR specifically provides a broad analytical coverage of metabolite species, ranging from aromatic compounds, amino acids, SCFAs, bile acids and gut microbial co-metabolites. Obviously NEN are bioactive tumours, and the secondary consequences of their metabolites are poorly defined.
- Patients were enrolled between 2011 and 2013, I wonder why data are submitted for publication in 2020
Complex and challenging, time consuming technology including meticulous quality control requiring substantial manpower on various levels, and the study design has contributed to the length of interval between the enrolment of the last patient and publication of data. Our aim was to perform our analysis in a cohort of patients with a largely accurate information about the biological behaviour of their tumour. Tumour grade (G1/G2) does not reflect the heterogeneity of the disease. We have therefore chosen a follow-up period of 36 months. This has also allowed us to gain information on potential predictive value of a 1HNMR based metabolic signature. Interestingly, all three PanNEN patients without distant metastases at the time of analysis that clustered close to spectra from patients with metastases on the negative side of principal component 1 developed recurrent disease during the 3 years follow-up. This highlights the potential for metabolic phenotyping to detect metastases earlier and more accurately than current clinical practice allows.
- I don't understand why patients were selected according with available follow-up > 3 years, I don't see data on patients' follow-up.
Our aim was to have a largely accurate information about the biological behaviour of their tumour. We have therefore chosen a follow-up period of 36 months. The follow-up protocol was according to recommendation of the European Neuroendocrine Tumour Society. This has also allowed us to gain information on potential predictive value of a 1HNMR based metabolic signature. Interestingly, all three PanNEN patients without distant metastases at the time of analysis that clustered close to spectra from patients with metastases on the negative side of principal component 1 developed recurrent disease during the 3 years follow-up (mentioned in the manuscript).This highlights the potential for metabolic phenotyping to detect metastases earlier and more accurately than current clinical practice allows. - There is no correlation with grading, staging, and validated tumor markers (this is also reported by the authors in the limitations of the study)
The sample size in the study is too low to allow for valid correlation with grading, staging, and standard tumour markers (chromogranin). This will be done in our future study on a higher number of patients.
- In the discussion, I believe a comment on other novel (and more solid) diagnostic tool should be given (i.e. NETest)
We have mentioned other novel omics-based biomarkers including the NETest in the introduction. We share the opinion of the reviewer that the NETest could be a valid biomarker in diagnosis, follow up and monitoring response to treatment of NEN. Since we are involved in studies assessing clinical utility of the NETest we wanted to avoid the impression of promoting it by mentioning it also in our non-NETest studies. We have now added a comment on it. - The number of healthy control is low, maybe it should be increased to better validate the model.
We appreciate this suggestion, however we would be reluctant to increase the number of healthy controls for this study and mix the new results with the existing. We would have to re-run all samples and re-analyse all data. We will however increase the number of healthy controls in our current ongoing study.
- Additional data on matching cases vs controls should be given (age, sex, drugs consumption).
Thank you for bringing this to our attention. Only individuals on no drug treatment were considered. We have added this information in the manuscript.
Reviewer 3 Report
This manuscript by Jimenez at al., performed the urinary metabolomic profiling of neuroendocrine neoplasms (NEN) patients by NMR and revealed dysregulations of several metabolites in NEN. The findings are informative and of value to the future development of biomarkers for NEN. The following comments serve to improve the present manuscript.
- Study subjects with high exclusion rate.
1) Among 175 Pan-NEN or SBNEN patients, 134 patients were excluded from this study. The exclusion criteria are: under the age of 18, pregnancy, patients undergoing NEN-specific systemic treatment, second malignancies, comorbidities requiring significant systemic treatment, impaired renal function, poor compliance, missing consent for study participation. This manuscript should reveal the exact patient numbers of each ineligible group, and explain why those patients were excluded with supportive references.
2) Among 41 eligible patients, 7 were further excluded because of poor water suppression (1 patient), high intensity of unknown drug signal (3 patients) and extreme misalignment of the spectra (3 patients). As the author mentioned that eligible NEN patients are limited, are there any technical procedures performed to improve the data quality?
- This manuscript didn’t reveal the total identified metabolites. Also, as day-to-day, person-to-person variations in the urine volume, a urinary metabolite such as creatinine is generally used to normalize samples. This manuscript didn’t specify how the normalization was performed.
- The details on the calculations for intra-assay coefficients of metabolite variations (CV%), and intra-individual and inter-individual metabolite variations are also lacking to show the robustness and accuracy of the methodology used in this manuscript.
- For OPLS-DA, a k-fold cross-validation was used to build up a model. The k=7 was used in this manuscript. Could the authors explain why k=7 was used, rather than k=5, the most common used?
Author Response
This manuscript by Jimenez at al., performed the urinary metabolomic profiling of neuroendocrine neoplasms (NEN) patients by NMR and revealed dysregulations of several metabolites in NEN. The findings are informative and of value to the future development of biomarkers for NEN. The following comments serve to improve the present manuscript.
We would like to thank the reviewer for his overall positive assessment of our research and his valuable comments.
- Study subjects with high exclusion rate.
1) Among 175 Pan-NEN or SBNEN patients, 134 patients were excluded from this study. The exclusion criteria are: under the age of 18, pregnancy, patients undergoing NEN-specific systemic treatment, second malignancies, comorbidities requiring significant systemic treatment, impaired renal function, poor compliance, missing consent for study participation. This manuscript should reveal the exact patient numbers of each ineligible group, and explain why those patients were excluded with supportive references.
Our inclusion criteria in this initial study on metabolic circuits in NEN were indeed very strict since we wanted to minimize cofounding variables and false discovery results as much as possible. We have also excluded those patients who already had their primary tumour resected and those who had any previous systemic, liver directed or surgical NEN-specific treatment. We have clarified this by adding additional information in the text. We respectfully disagree with the reviewer that the exact number of patients in each ineligible group would be of interest for the readership. Moreover, since many patients had various treatment modalities, either as consecutive single measures or in combination, during course of the disease, the numbers would cause confusion. To the best of our knowledge exact patient numbers of each ineligible group at entry or any data of patient population not studied are not required in other studies.
We are aware that the scenario we have considered does not reflect the real word experience, however strict selection was inevitable for an analysis of base line metabolic characteristics of NEN and insight into disease pathophysiology. Since treatment has a dramatic effect on metabotypes, analysing treatment naïve individuals is crucial for a discovery study (Berger RD et al. Metabolomics 2016, Nicholson JK et al. Nature 2012, Kinross J et al. Sci Rep 2017).). The principle of inclusion of only treatment naïve patients is followed also by others; e.g. in a study on NMR spectroscopy of filtered serum of prostate cancer patients only patients who have not been administered any treatment or suffered any comorbid milieus were included (Kumar D et al. Prostate 2016) and in a study on metabolomic profiling in patients with endometrial cancer all patients who had any type of oncologic treatment [surgery, chemotherapy, radiotherapy] were excluded (Kozar N et al. Adv Med Sci 2020).
2) Among 41 eligible patients, 7 were further excluded because of poor water suppression (1 patient), high intensity of unknown drug signal (3 patients) and extreme misalignment of the spectra (3 patients). As the author mentioned that eligible NEN patients are limited, are there any technical procedures performed to improve the data quality?
Poor quality spectra are re-analysed at the time of acquisition, and analytical technical failures are discarded as part of the quality control process. However, it is also possible that sample characteristics can be the cause of low-quality analytical data results, e.g. sample dilution causes poor water suppression; intense signals belonging to exogenous metabolites such as drugs can also distort the base line, or cause misalignment of signals belonging to other metabolites in the 1H NMR spectrum. Analytical data obtained from patient samples are therefore subject to strict quality control assessment in order to prevent bias in the statistical models. We have referenced these processes in the paper (reference # 49)
- This manuscript didn’t reveal the total identified metabolites. Also, as day-to-day, person-to-person variations in the urine volume, a urinary metabolite such as creatinine is generally used to normalize samples. This manuscript didn’t specify how the normalization was performed.
The total number of identified metabolites was 250. Only metabolites with discriminatory value were used for further analysis. Our in-house data base and the HMDB data bank (Human Metabolome Data Base http://www.hmdb.ca) were used to verify the characteristic resonances. We have added this information in the manuscript.
The probabilistic quotient normalization based on the calculation of a most probable dilution factor by looking at the distribution of the quotients of the amplitudes of a test spectrum by those of a reference spectrum was used. The type of normalisation applied is now described in the text in addition to the reference #54. Probabilistic quotient normalization (PQN) is the optimal method for normalising 1H NMR data from samples such as urine because it accounts for dilution factors and avoids bias introduced by the presence of metabolites found in high concentrations in specific samples. The concentration of creatinine in urine is not a constant value, because it is subject to variation caused by sample dilution, dietary habits, BMI and other biological factors.
- The details on the calculations for intra-assay coefficients of metabolite variations (CV%), and intra-individual and inter-individual metabolite variations are also lacking to show the robustness and accuracy of the methodology used in this manuscript.
CV% are not reported in the text. However, correlation coefficients and p values are presented in figure 4 and table 2. P values are accepted statistical parameters used to assess whether the differences observed in the concentration of a metabolite in two different cohorts is statistically different or not. Our work presents evidence of several metabolites being significantly different in concentration between NEN patients and healthy controls but also between the urinary metabolic profiles of NEN patients with different primary tumour sites.
- For OPLS-DA, a k-fold cross-validation was used to build up a model. The k=7 was used in this manuscript. Could the authors explain why k=7 was used, rather than k=5, the most common used?
k=7 leave one out cross validation is typically used when performing metabolic profiling analysis with 1H NMR spectroscopical data. As stated in “Applied Predictive Modelling” by Max Kuhn and Kjell Johnson (DOI 10.1007/978-1-4614-6849-3e) the choice of k is usually 5 or 10, but there is no formal rule. As k gets larger, the difference in size between the training set and the resampling subsets gets smaller. As this difference decreases, the bias of the technique becomes smaller”. We have therefore used k=7 as it provided the optimal analysis in this data set.
Reviewer 4 Report
I congratulate the authors of this study that has the merit of analyzing the intriguing diagnostic potential of the metabolomics approach in pancreatic and small bowel well-differentiated neuroendocrine neoplasms.
In this prospective controlled observational study, the authors investigate the diagnostic utility of urine metabolomics and the diagnostic role of 1H-NMR for NEN collecting urine samples of 34 untreated G1-G2 NEN patients, of which 7 were functioning and 14 presented with liver metastases.
Important drawbacks of this study were small numbers of cases and a high selection of patients, which are properly discussed by the authors.
We suggest the following modifications:
- Limitations concerning the diagnostic utility of standard serum tumour markers should be expounded on a bit more in the discussion.
- Page 11, lines 199-200: “Liao et al., 199 demonstrated that trigonelline”. Please, correct the format character.
Author Response
I congratulate the authors of this study that has the merit of analyzing the intriguing diagnostic potential of the metabolomics approach in pancreatic and small bowel well-differentiated neuroendocrine neoplasms.
In this prospective controlled observational study, the authors investigate the diagnostic utility of urine metabolomics and the diagnostic role of 1H-NMR for NEN collecting urine samples of 34 untreated G1-G2 NEN patients, of which 7 were functioning and 14 presented with liver metastases.
Important drawbacks of this study were small numbers of cases and a high selection of patients, which are properly discussed by the authors.
We are appreciating the reviewer’s overall positive assessment of our work and we would like to thank for his suggestions how to improve the quality of the manuscript.
We suggest the following modifications:
- Limitations concerning the diagnostic utility of standard serum tumour markers should be expounded on a bit more in the discussion.
We had added information on chromogranin A as a standard blood tumour marker for NEN in the text - Page 11, lines 199-200: “Liao et al., 199 demonstrated that trigonelline”. Please, correct the format character.
Thank you for bringing this to our attention. A correction has been made.
Round 2
Reviewer 2 Report
I thank the authors for their extensive reply. In my opinion, some of the major concerns raised by the initial evaluation remain unaddressed.